# On the optimality of 2°C targets and a decomposition of uncertainty

Kaj-Ivar van der Wijst [1,2 ✉], Andries F. Hof [1,2] & Detlef P. van Vuuren [1,2]

Determining international climate mitigation response strategies is a complex task. Integrated Assessment Models support this process by analysing the interplay of the most relevant factors, including socio-economic developments, climate system uncertainty, damage estimates, mitigation costs and discount rates. Here, we develop a meta-model that disentangles the uncertainties of these factors using full literature ranges. This model allows comparing insights of the cost-minimising and cost-benefit modelling communities. Typically, mitigation scenarios focus on minimum-cost pathways achieving the Paris Agreement without accounting for damages; our analysis shows doing so could double the initial carbon price. In a full cost-benefit setting, we show that the optimal temperature target does not exceed 2.5 °C when considering medium damages and low discount rates, even with high mitigation costs. With low mitigation costs, optimal temperature change drops to 1.5 °C or less. The most important factor determining the optimal temperature is the damage function, accounting for 50% of the uncertainty.

[1] PBL Netherlands Environmental Assessment Agency, The Hague, The Netherlands. [2] Copernicus Institute of Sustainable Development, Utrecht University, Utrecht, The Netherlands. ✉email: kajivar.vanderwijst@pbl.nl

A s part of the United Nations Framework Convention on Climate Change (UNFCCC), countries have agreed to prevent 'dangerous anthropogenic' climate change. In the Paris Agreement, this was specified further as the aim to keep the increase of global mean temperature change well below 2°C and pursuing efforts to limit it to 1.5°C. Determining a goal for international climate policy is extremely complex, as it involves many socio-economic, geophysical and even ethical aspects. To explore and understand this complexity, researchers have developed Integrated Assessment Models (IAMs) describing the interplay of several factors relevant to climate change.

A plethora of IAMs has already been developed, with varying degrees of complexity and differing in focus. One category of models focuses on cost-minimising carbon price or emission pathways to achieve a specific climate target[1–5]. A second category consists of models that determine optimal pathways, which balance the costs and benefits of climate policy[6–10]. In this type of models, the climate target is an outcome rather than determined exogenously. These two types of models have developed relatively independently. However, in both types, a (shadow) carbon price is used as a key indicator of mitigation effort and costs associated with the transition towards a low-carbon future—and the development of carbon prices and emissions forms a key component in both types of models.

Several studies have analysed the effect of various assumptions and uncertainties (for instance, related to the discount rate, climate sensitivity or the damages of climate change) on the optimal pathway. However, such studies are often limited in scope[11,12], only perform a sensitivity analysis[13,14], do not capture the latest insights (e.g., outdated damage functions)[15–17], or perform a simulation instead of an optimisation[17,18]. Moreover, no studies exist that have compared cost-minimising pathways with cost-benefit pathways using the same model framework—except for Nordhaus[19], who did this for a few selected assumptions regarding discounting and climate targets. Such a comparison would provide insight into under which conditions taking into account climate damages would change the cost-optimal carbon price and emission pathway, given a fixed climate target.

A comprehensive analysis of cost-benefit versus cost-minimising pathways, including an uncertainty analysis of the most important parameters, requires a model that is simple enough to use mathematical optimal control theory techniques but complex enough to capture the relevant technological and socio-economic dynamics. Moreover, the model should easily be calibrated to the literature ranges. In this paper, we develop a flexible and transparent model to calculate the optimal carbon price path under a set of assumptions regarding damage functions, temperature goals, mitigation costs, climate sensitivities, discount rates and socio-economic developments. With this model, we directly compare the insights of the two main Integrated Assessment Modelling communities: the cost-minimising models which focus on how climate targets (e.g., a carbon budget) can be reached, without taking damages into account, and the cost-benefit models which compare the marginal mitigation costs to marginal damages to calculate optimal temperature goals.

With this model, we first analyse each parameter's effect on the timing of mitigation in cost-minimising paths (also called cost-effective paths). We then quantify how these cost-minimising mitigation paths are impacted if the economic impact of climate damages is included, and not only mitigation costs. We analyse how the relative importance of each parameter's uncertainty varies over time.

Besides cost-minimising paths with a carbon budget, we analyse optimal cost-benefit paths (which do not require a preset carbon budget). In particular, the resulting optimal end-of-century temperature has been the subject of much research. Here, we provide a comprehensive analysis of how this optimal temperature depends on the literature ranges of the relevant parameters—moving beyond current literature that only considers a limited range of damages or mitigation costs[13]—and investigate under which assumptions the 2°C temperature target set by the Paris Agreement is optimal.

We move beyond studies presenting sensitivity analysis of the assessed parameters and conduct a systematic uncertainty analysis using ranges based on literature. We also analyse the interaction between parameters and assess to which degree uncertainty in individual parameters affect total uncertainty in the optimal carbon price or end-of-century temperature.

The model used in this paper is based on a simple economic growth model (Fig. 1). This model shows some similarities with the DICE[8] and the FAIR[15] model. The production function is combined with estimates on mitigation costs and climate damages from recent literature. In the model, a global carbon price is applied such that the discounted utility is maximised. This transparent model is still solvable using the Bellman equation, which guarantees mathematical optimal solutions.

The model is calibrated using literature ranges on parameters relevant for global climate policy (highlighted in colour in Fig. 1). The socio-economic variables are obtained from the Shared Socio-economic Pathways (SSPs, blue)[20]. The damage functions (green) cover the low range of damage functions (DICE 2016R2-damage function[21]), the medium (based on a meta-analysis by Howard et al.[22] of empirical and traditional IAM estimates and referred to as Howard Total in this article), and the high range (long-run empirical damage function from Burke, Hsiang and Miguel[23]). Both the Transient Climate Response to Emissions (TCRE, pink), linking temperature to cumulative $CO_2$ emissions, and the mitigation costs (yellow), are calibrated to IPCC AR5 data[24,25] and both span the 5–95th percentile range[26]. Finally, we use three values for the pure rate of time preference (purple): 0.1%, 1.5%, and 3% per year. The values used for each parameter are summarised in Table 1. Although other parameters like different technological growth assumptions, social inertia and welfare are relevant, their impact on this paper's main policy outcomes is significantly smaller than the five main parameters we focus on in this paper (see Discussion).

With our model, we discuss how these parameters affect optimal carbon price paths and associated emission paths in a cost-minimising setting, by imposing a carbon budget. When considering cost-minimising pathways reaching the Paris Agreement's temperature target, including medium damages can double the initial carbon price compared to purely considering mitigation costs. Moreover, decreasing the pure rate of time preference from 1.5% to 0.1% also doubles the initial carbon price. Over the century, the cost-minimising carbon price mostly rises with per capita GDP growth. The level of mitigation costs dominates the variance of the carbon price. The discount rate, damage function and socio-economic scenario contribute in almost equal part to the remaining variance, with a drop in absolute variance around 2070. Consequently, the choice of discount rate and how climate damages are valued have a substantial effect on the carbon price in a cost-minimising setting. To reduce the uncertainty in climate policy, these choices have to be made as soon as possible.

In a cost-benefit setting (without carbon budget), even with high mitigation costs, the optimal end-of-century temperature with medium damages and a low discount rate does not exceed 2.5°C. For low mitigation costs or with the high damage function, we find an optimal temperature of 1.5°C or less. The effect of a different TCRE is negligible for scenarios with an optimal temperature between 1.5 and 2°C. Over 50% of the uncertainty comes from the damage function, compared to only 2% from the TCRE.

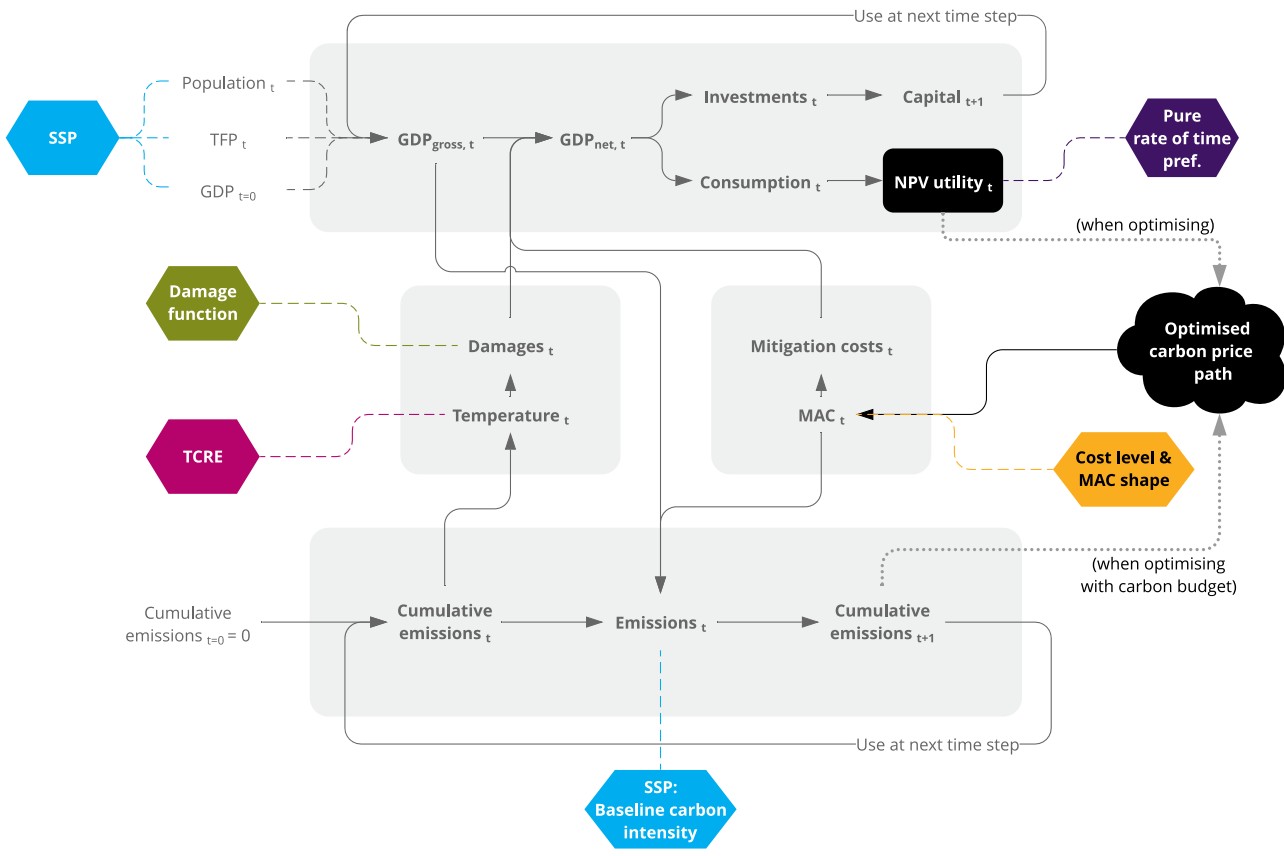

**Fig. 1 Schematic representation of the model.** The model consists of an economic module (top) with a Cobb-Douglas production function, and an emission module (bottom). The interactions between these two modules occur through damages and mitigation costs. The coloured boxes represent the parameters for which we use a representative range from literature. The carbon price path, in black, is the input variable of the model and the control variable in the optimisation.

| Table 1 Values for the main parameters of the model. | | | | | |
|---|---|---|---|---|---|
| **Parameter** | **SSP** | **Damage function** | **TCRE** | **Mitigation cost level** | **Pure rate of time preference** |
| **Values** | SSP1, SSP2*, SSP3, SSP4, SSP5 | No damage**, DICE 2016R2 (low), Howard Total (middle)*, Burke (LR) (high) | 0.42, 0.62*, 0.82 °C/1000 GtCO$_2$ | From IPCC AR5 consumption losses: low, medium* and high | 0.1%/yr, 1.5%/yr*, 3.0%/yr |

*Default parameter value if not specified.
**Only used in cost-minimising scenarios with a fixed carbon budget.

Many of these results individually are consistent with previous research. This paper presents a comprehensive overview of the relative importance of each of these results.

## Results

**Optimal carbon price paths with a fixed carbon budget.** This section focuses on optimal carbon price paths reaching a fixed carbon budget in 2100 (cost-minimising setting). The carbon budget here is 1344 GtCO$_2$, which leads to a 2°C temperature increase compared to pre-industrial with 67% certainty given the normal distribution of the TCRE (see SI. 4.1). The effect of changing various parameters on the shape of the carbon price and, subsequently, the emission path, is shown in Fig. 2. Each experiment compares the cost-minimising path without damages (solid lines) with cost-minimising paths including damage costs, based on the medium damage function Howard Total (dotted

lines). Moreover, the effect of varying each remaining parameter individually (mitigation cost level, SSP, TCRE and discount rate) is analysed (various colours in each subplot). In each of the experiments, we use the default values of Table 1, unless specified otherwise.

In each of the experiments, the carbon prices increase over time, before they start falling again when the imposed minimum emission level, set to represent restrictions on carbon dioxide removal technologies (see Methods), is reached (if at all). Including damages leads to a shift of the mitigation effort from the end of the century towards the present: the optimisation aims to reduce the impact of damages on GDP development by increasing the mitigation effort early on. Consequently, the carbon price path becomes more linear. This impact depends on the damage function and is smaller for the DICE damage function and strongest for the Burke function. In fact, in the latter case, the optimisation can lead to smaller carbon budgets than the target.

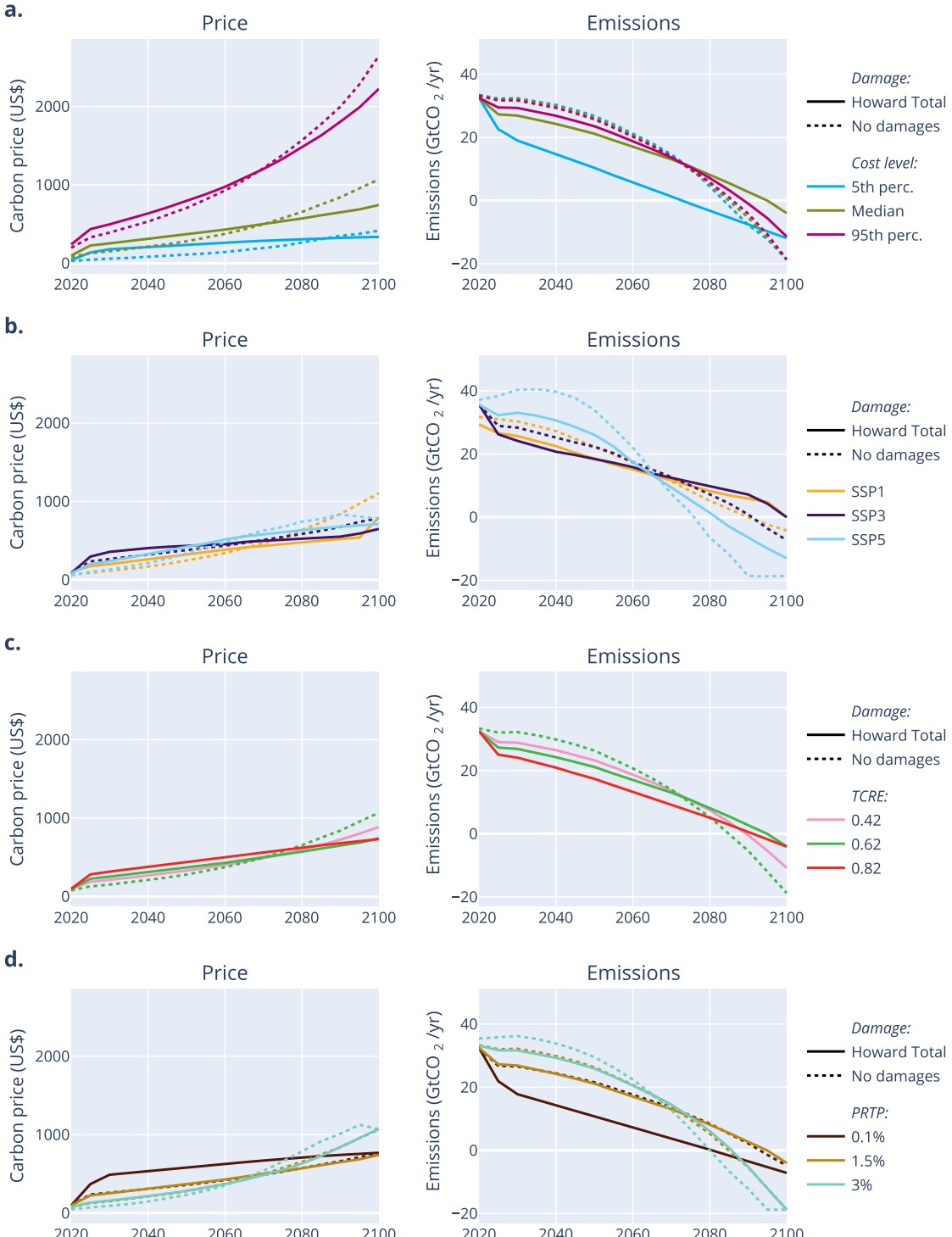

**Fig. 2 Optimal carbon price paths (left) with corresponding emission path (right) for different scenarios with a 1344 GtCO₂ carbon budget (cost-minimising setting).** For each scenario, the default parameters (see main text) are used, with one parameter changed (**a** mitigation cost level, **b** SSP, **c** TCRE, **d** pure rate of time preference). The solid lines correspond to purely cost-minimising paths (no damages), the dotted lines take into account the medium damage function Howard Total.

Unsurprisingly, higher marginal abatement costs lead to higher carbon prices to reach the given carbon budget (Fig. 2a). In the no-damage scenario, the carbon price path is linearly dependent on the height of the MAC and so there is no impact on timing (Supplementary Figure 6.1). Interestingly, when including damages, the initial carbon price (in 2025 to avoid initial inertia

constraints) depends on the interaction between damages and mitigation costs. For high mitigation costs, the medium damage function implies an initial price that is 32% higher than without taking damages into account, whereas for low mitigation costs, this is 282% higher. When using the higher (Burke) and lower (DICE) damage functions, this effect also exists but is larger or

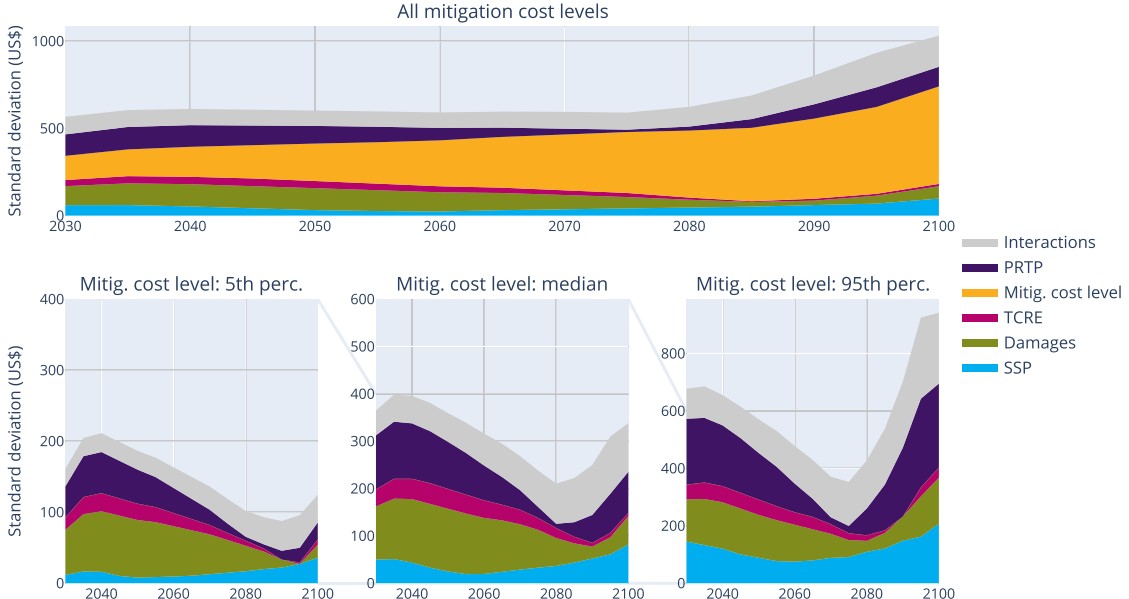

**Fig. 3 Contribution to the variance of each parameter as a function of time using Sobol Indices (cost-minimising setting).** In the top row, all parameters are considered. In contrast, in the bottom row, the same analysis is performed while fixing the mitigation costs at three distinct levels: low, medium and high costs. Note that for clarity, the square root of the variance—the standard deviation—has been shown: the unit then becomes US$ instead of the square of it. The decomposition with variances is shown in Supplementary Figure 2.2.

smaller, respectively. When assuming medium damages, taking into account damages to determine the optimal emission pathway only leads to a substantially different optimal emission pathway if low mitigation costs are assumed, with very early reductions and hardly any net negative $CO_2$ emissions.

The SSP substantially impacts the optimal carbon price (Fig. 2b) (see also[20]). First of all, the difference in baseline emissions (e.g., between SSP1 and SSP5) explains that the SSP5 carbon price path grows earlier and more rapidly. Despite this higher carbon price, the minimum emission level is reached before the end of the century in SSP5. Second, since utility is derived from per capita consumption, a high population growth combined with a low GDP, as in SSP3 (Supplementary Figure 1.2), means that end-of-century costs have a larger impact on total cumulative utility. Therefore, the mitigation effort is more linear in SSP3: a much higher initial carbon price is followed by lower carbon prices towards 2100 compared to the SSP1 and SSP5 paths. In other words, cost-minimising paths without damages lead to initially exponentially increasing price paths, unless one assumes that our future society will not be much richer than today, confirming previous findings[27].

The TCRE dependence (Fig. 2c) is relatively straightforward: the higher the TCRE, the higher the impact of damages on the carbon price and emission pathway (thus leading to a stronger preference for early mitigation). In fact, the initial carbon price increases almost linearly with the TCRE (Supplementary Figure 6.3).

Higher discount rates shift mitigation efforts towards the future (Fig. 2d). In a cost-minimising setting without damages, decreasing the pure rate of time preference from 3% to 1.5% almost doubles the initial carbon price. Moving from 1.5% to 0.1% almost doubles the initial price again.

Subsequently, we analyse the combined effect of all parameters on the optimal carbon price and determine each parameter's contribution to the total variance. This is quantified by Sobol indices, calculated with a Monte Carlo simulation using each combination of parameter values of Table 1 (see Methods): the total variance is split in partial variances attributed to each parameter, along with interactions between them. As we consider scenarios with a fixed carbon budget here, we focus on the determinants for the optimal carbon price only. The top panel of Fig. 3 shows the standard deviation of the optimal carbon price and its determinants over time. The standard deviation remains relatively constant until the mid-2060s, after which it increases strongly, as a result of the increasing mean values of all carbon prices over time. The dip in variance in the first decade comes from the constraining effect of inertia to reduce initial emissions. The main contribution to the variance is by far the mitigation cost level, especially in the longer term. However, the initial carbon price is also strongly influenced by future socio-economic developments (through the SSPs, see also Supplementary Figure 2.4). To better analyse the contribution of the remaining parameters, we perform the same analysis but by fixing the cost level at low, medium and high mitigation costs (bottom three panels of Fig. 3).

The variance of the optimal carbon price due to other determinants is the highest towards the end of the century. Interestingly, for all cases presented there is very little variance around 2070. This can be explained by the fact that most changes in parameter values induce a shift in mitigation effort either towards the present or towards the end of the century. They therefore increase (or decrease) the initial carbon price, and decrease (or increase) the final carbon price—leading to similar carbon prices by 2070.

The SSP, discount rate and damage function contribute equally to the total variance for medium mitigation cost levels. For low mitigation costs, the damages become more important. In contrast, the SSP becomes more dominant for high mitigation cost levels, where the marginal mitigation costs become substantially larger than the marginal damages. The contribution of the uncertainty in TCRE is negligible, accounting for <0.5% of the variance. This confirms previous findings[26], which state that the socio-economic uncertainty is far more important than the geophysical uncertainty in scenarios with stringent temperature targets.

For all the cost-minimising 2°C pathways, it can be determined whether the monetary benefits (damages avoided compared to a

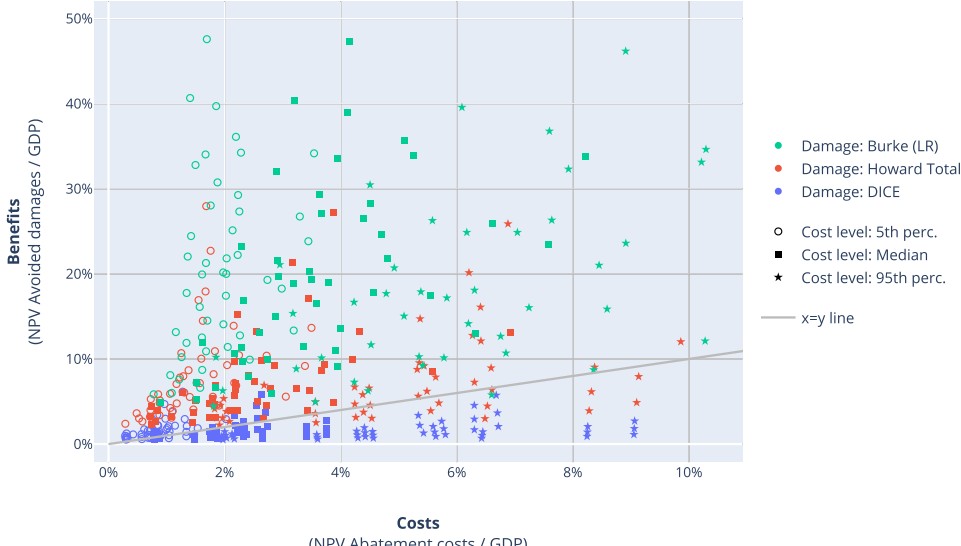

**Fig. 4 Costs versus benefits (cost-minimising setting).** The costs are calculated as the net present value (NPV) of abatement costs as a share of GDP, whereas the benefits are the NPV of avoided damages as a share of GDP compared to the baseline SSP scenario, for each scenario reaching the carbon budget of 1344 GtCO2, and for each combination of parameters of Table 1.

baseline scenario) outweigh the costs (net present value of abatement costs). Although this does not imply that the pathways are optimal in a cost-benefit setting, at least the pathways lead to net benefits compared to baseline if this is the case. This comparison is very similar to the comparison of the Stern Review[28], in which a stringent scenario was compared to no mitigation at all. Of all parameter combinations with either medium or high damages, 95% lead to avoided damages exceeding mitigation costs (Fig. 4). The remaining 5% consists mostly of high mitigation cost scenarios. For the DICE damage function, only 40% of all parameter combinations lead to higher benefits than costs. The magnitude of the damages, and much less the magnitude of the mitigation costs or the discount rate, therefore largely determines whether the benefits of 2°C outweigh the costs. The optimal balance between avoided losses and mitigation costs—the cost-benefit setting—is another question, however, and is discussed in the next section.

**Cost-benefit paths (without a carbon budget).** This section considers purely cost-benefit scenarios, without carbon budget or temperature target: the optimal price path results from an optimal balance between mitigation costs and damages. We discuss the optimal temperature in 2100 for different parameter combinations and subsequently, we analyse the contribution to the variance of the optimal temperature resulting from each parameter. Finally, we briefly discuss the resulting shape of the optimal carbon price path in a cost-benefit setting.

Figure 5 shows the optimal temperature in 2100 for all combinations of the discount rate, damage function, mitigation cost level, and SSP. A 2°C temperature target or lower is found to be optimal in most parameter combinations in cost-benefit settings with high damages or with a low discount rate, the latter with the exception when combined with DICE damages. Low discounting does not always lead to optimal temperatures below 2°C, especially if high mitigation costs and medium to low damages are assumed. However, in most cases, the optimal temperature is 3°C or significantly less with low discounting, except for SSP5 socio-economic developments. The optimal temperature in SSP5 is consistently higher than in other SSPs: between 3°C and 4.5°C. On the other hand, SSP1 and SSP4 have consistently lower optimal temperatures (between 1°C and 3°C), directly correlated with the

baseline emissions of these SSPs. In fact, with a high discount rate or low damages, the influence of the SSP becomes much more important. This is discussed in more detail below.

The effect of assuming a low or high TCRE instead of the median value is mostly linear with the optimal temperature (Supplementary Figure 3.4): the higher the optimal temperature with median TCRE, the higher the effect. A lower TCRE leads to a lower optimal temperature. Conversely, a high TCRE leads to a higher optimal temperature, but this effect is dampened by an increased abatement effort to counter the increased damages. Scenarios with an optimal temperature around 2°C hardly see any impact of a change in TCRE.

To assess the contribution to the variance of each model parameter on the optimal temperature, we perform a Sobol variance decomposition using the same method as for the variance decomposition of the carbon price in a cost-minimising setting. The difference here is that we focus on the optimal temperature in 2100 instead of the carbon price over time. The total variance is split into percentages attributed to each parameter. When considering all combinations of parameter values, the damages are responsible for the highest variance (58%), followed by the discount rate (15%) and the mitigation cost level (14%). This is shown as the central node in the conditional variance tree of Fig. 6. We split on the variable with the highest variance (damages) and perform the same analysis, conditional on each value of this parameter. By repeating this process, we fix the parameters' values with the highest conditional variance and obtain a tree structure.

Interestingly, the parameters with the highest variance within each level of this tree (large grey circles in Fig. 6) are not identical. For instance, with a medium or high damage function (Howard Total and Burke), the discount rate and mitigation cost level dominate the variance since the mitigation effort level is then mainly determined by how much weight is given to costs for future generations. When considering a low damage function (DICE) with a high discount rate, where the optimal emission path is closer to the baseline emissions, the next parameter with the highest variance is the SSP. The mitigation costs in this case only play a significant role in SSP5 (with its higher baseline emissions), whereas, for the other SSPs, the geophysical uncertainty in TCRE dominates the variance.

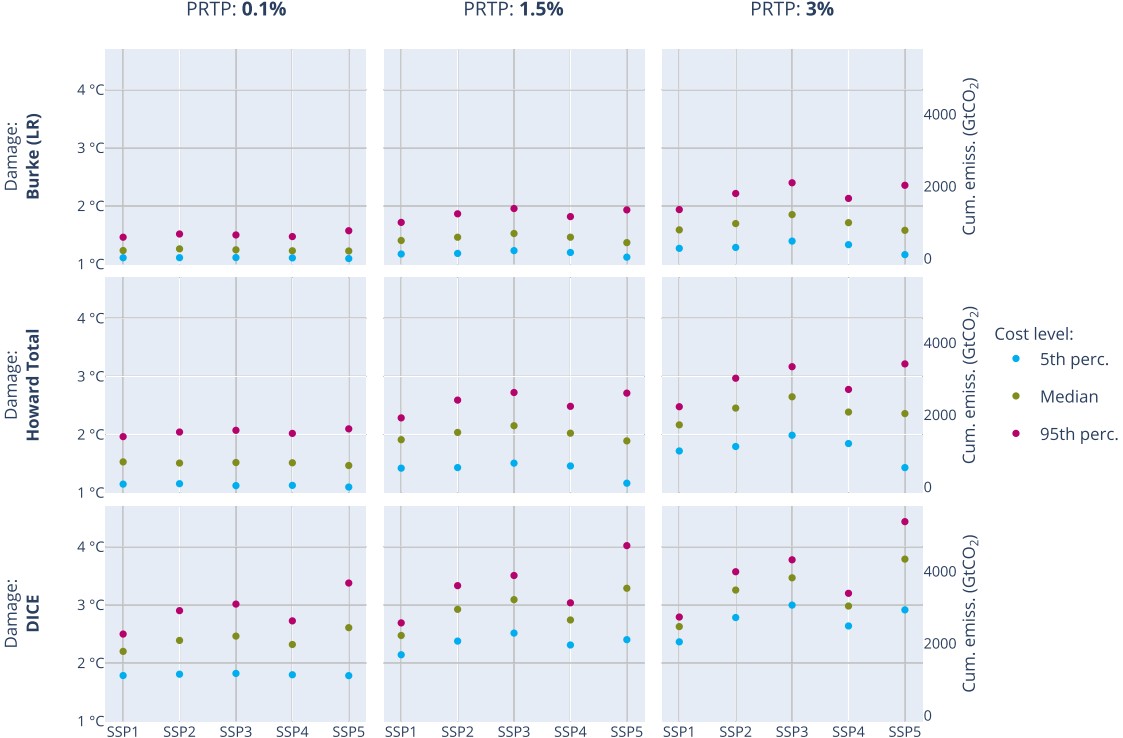

**Fig. 5 Optimal temperature in 2100 (cost-benefit setting).** The optimal end-of-century temperature is shown for three different pure rate of time preference rates (columns), three damage functions (rows) and mitigation cost levels (colours). The median value of the TCRE is used for each scenario here. Therefore, the end-of-century temperature corresponds linearly to the cumulative $CO_2$ emissions from 2020 to 2100.

The interaction terms, represented in grey in Fig. 6, can be further decomposed. As shown in Supplementary Fig. 3.8, the two highest interaction terms are between SSP and the damage function (owing to the large differences in baseline between SSPs, where some SSPs are much more sensitive to climate damages) and between the TCRE and the damage function (since the TCRE has a direct influence on the temperature and therefore the damages). This shows that although the first-order variances of SSP and TCRE are small, their total variance is larger when including interaction terms.

The timing of mitigation is obviously at least as important as the optimal temperature. The optimal carbon price paths in a purely cost-benefit setting increase almost linearly (Supplementary Figure 3.3), consistent with simpler settings in earlier studies[27]. Although greater damages lead to slightly steeper carbon price paths, most factors influence mainly the initial carbon price. The exception here is the SSP (panel b, Supp. Fig. 3.3): the SSP determines mostly the steepness of the carbon price path. Similar to cost-minimising paths, the SSP3 path is much flatter than the other SSPs. The uncertainty in carbon prices in the cost-benefit setting (sometimes called the social cost of carbon) can be directly compared to the carbon price's corresponding uncertainty in the cost-minimising setting (cf Fig. 3 and Suppl. Fig 3.4b). Interestingly, uncertainty in mitigation costs has a much smaller impact on the level of the cost-benefit carbon price (Supplementary Figure 3.4 a and b): it explains ~10% of the total variance. The damage function (45–60%) dominates the variance instead. The discount rate is the most important factor for low damages, whereas conditional on high damages, the mitigation cost level contributes most to the variance.

## Discussion

This paper focuses on the economic aspects of climate policy by discussing cost-minimising paths and optimal temperatures in a cost-benefit setting. The approach provides insight into the critical factors that determine the attractiveness of various mitigation pathways. Moreover, it allows extending on current literature research on cost-benefit analysis.

This paper also adds some nuance to the claims of recent literature[12,13] stating that the 2°C temperature goal, as set in the Paris agreement, is indeed optimal. Glanemann et al. (2020) focus on the Burke baseline damage functions (short run). This damage function corresponds roughly to the Howard Total damage: in our analysis, the optimal temperature in 2100 using Burke (short run) divided by the results using Howard Total damages has a mean of 0.97 (standard deviation of 0.07). Although the optimal temperature using medium mitigation costs and a small discount rate is indeed very close to 2°C, different mitigation cost levels or discount rates have a strong impact, leading to optimal temperatures between 1.1 and 3.5°C. This confirms the importance of considering the full literature range for these parameters. On the other hand, Glanemann et al. observe a larger impact of using different climate sensitivities (moving the optimal temperature from 2°C to 1.5 or 2.5°C for different climate sensitivities). This difference is likely owing to DICE's different climate module: whereas we use the instantaneous TCRE relation, DICE uses a two-box model with much longer lag times. Similarly, our results are in line with Hänsel et al.[12], considering that they used a similar range in social discounting parameters, but only our medium estimates for climate damage and mitigation costs. Our full range of optimal climate targets is much larger. Whereas Drouet et al.[17] use emission pathways generated using more detailed IAMs, the damages were only added afterwards. Considering the climate damages in the optimisation leads to significant differences in carbon prices, as shown in Fig. 2. Moreover, the optimal carbon budget candidates selected in ref. [17] are higher than our optimal carbon budgets, mainly due to the much lower, now outdated, damage functions employed in their

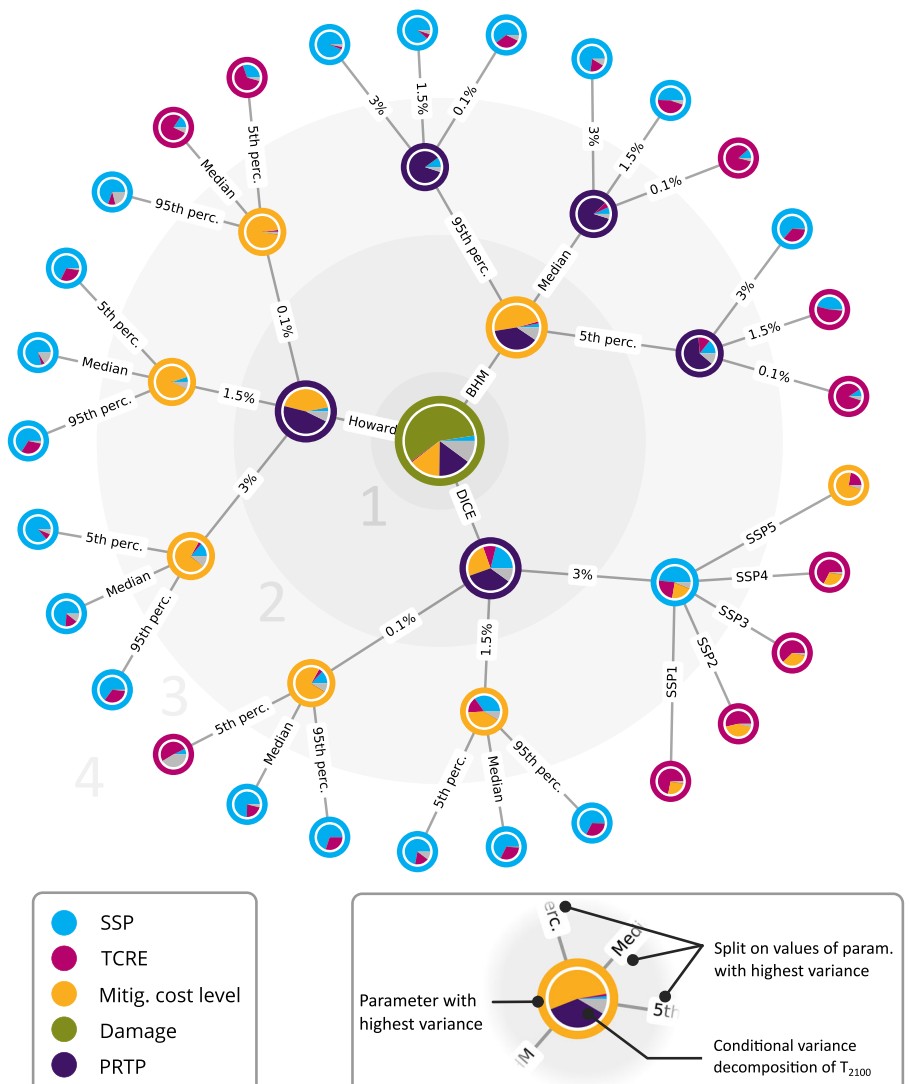

**Fig. 6 Conditional variance tree for the temperature in 2100 (cost-benefit setting).** At the central node, a Sobol variance decomposition is performed on the whole set of parameter values. The pie chart represents the percentage each parameter contributes to the total variance. The outer colour is the parameter with highest variance. The node is split in each of this parameter value, and the variance decomposition is repeated with this parameter value fixed. By repeating this process, a conditional variance tree is created. The grey colour in each node represents the interaction terms in the Sobol decomposition.

paper. Finally, our uncertainty decomposition contrasts with Lamontagne et al.[18], as we show a much larger uncertainty from damage function and mitigation costs. This difference is directly attributable to the use of full literature ranges instead of much smaller damage and abatement cost sensitivity ranges in ref. [18].

**Sensitivity runs**. Moving from a quadratic MAC to a cubic MAC has a small effect on the optimal temperatures (Suppl. Figure 5.2): more cheap mitigation options are available with a cubic MAC, but options become quickly expensive after these cheap options have been implemented. This leads to a smaller spread in optimal temperatures. In the cost-benefit setting, using a cubic MAC only significantly affects the carbon price in the low mitigation cost scenario (Supplementary Figure 5.3), where the carbon price is increased by ~20%, necessary to reach the more expensive high mitigation options of the cubic MAC. On the other hand, the cubic MAC only has a significant effect on the carbon prices in a cost-minimising setting when assuming high mitigation costs: to reach the 2°C target, very high carbon prices are warranted with

high mitigation costs, where the difference between the MACs is highest (Supplementary Figure 5.1).

As an alternative welfare formulation, we have performed the same analysis with the PRTP and elasticity of marginal utility (elasmu) values from a recent expert elicitation[29]. Using 172 combinations of these parameter values, the 5th, 50th and 95th percentile values can be calculated (see SI.7), giving a significantly wider range in the social discount rate. In fact, instead of using the PRTP values of 0.1%, 1.5% and 3% with a fixed elasmu of 1.001, the Drupp et al. (2018) survey yields 0% and 0.5, 0% and 1.5 and 2% and 2.5 for the PRTP and elasmu values, respectively. However, as these values can not be considered to be uniformly distributed (as in our main analysis), the effect on the uncertainty is very small (Supplementary Figure 7.3 and 7.6). For the main analysis, we have chosen to use the literature range of PRTP values instead of focusing on single expert elicitation.

Changing the minimum emission level from −20 GtCO₂ to 0 (therefore, avoiding an emission/temperature overshoot) influences the results in varying ways (SI 5.2). In the carbon budget setting, the mitigation effort is shifted towards the first half of the

century owing to the extra constraint. With medium assumptions and no damages, this leads to an 18% higher carbon price. Assuming greater damages reduces this difference since these scenarios already used less net negative emissions. In the cost-benefit setting, the impact of avoiding overshoot is negligible for scenarios ending up above 2°C, since the optimal emission paths leading to these temperatures hardly use net negative emissions in the first place. For lower optimal temperatures, the constraint leads to an increase in end-of-century temperature up to 0.2°C in SSP2 and up to 0.3°C in SSP5.

Many factors are not captured in the current model and therefore insights in general outcomes are more interesting than the absolute numbers. Key factors not included are, for instance, heterogenous impacts (for different societal groups) and the possibility of environmental feedbacks and tipping points, possibly with stochastic behaviour. We have chosen the Howard Total damage function to account for the missing tipping point modelling, which includes catastrophic damages through a proxy for tipping points in a traditional IAM damage function. Other factors like inequality and regional heterogeneity cannot be addressed with our global model—moving to a regional model would provide further insights.

In this research, we have considered a large range of damage functions, implicitly considering a wide array of assumptions on climate impacts. In future work, it would be interesting to disentangle this uncertainty. For example, recent work has shown that the role of biodiversity and ecosystem services and the associated scarcity of environmental goods is relevant for cost-benefit work[30]. Including natural capital in the production function would be a first step towards decomposing the damage uncertainty. Using bottom–up sectoral climate damages could decompose this further.

**Parameter validity**. The Burke damage function (Supplementary Figure 1.4) as calculated using our calibration reports end-of-century damages that are significantly lower than the damage function shown in Burke et al. (Extended Data Fig. 6)[23]. This difference is due to a combination of three factors: (1) damages to the GDP also affect future GDP growth due to a loss in capital, (2) Burke et al. assume a linear increase in temperature instead of the baseline SSP temperature increase and (3) the global estimates are slightly lower than the sum of local estimates owing to downscaling factors and the non-linearity of the temperature-growth impact relationship.

**Uncertainty**. Throughout this paper, we have considered the extensive range of key parameter values as the source of uncertainty. However, these represent the fact that these parameter's precise values are unknown—and not the uncertainty in a stochastic sense. Adding stochasticity to the model would allow for a more comprehensive investigation of the impact of tipping points. Previous work on stochastic IAMs[31,32] could be extended to include the literature ranges of the parameters provided in this paper, and possibly be extended with data on socio-economic tipping points[33]. This would effectively widen the range of possible damages.

**Suitability of the model**. The use of IAMs, in general, has been criticised both for cost-benefit analysis (for models like DICE)[16] and cost-minimising analysis, but this general critique has been discussed elsewhere[34]. Our analysis' added value is that it allows us to investigate the effect of critical, normative assumptions on policy-relevant quantities, like the magnitude of the carbon price or the optimal temperature. The simple model allows for more transparency in how our results are obtained and our parameters

are calibrated. Moreover, some of the criticisms, such as ad hoc input parameters and damages[16], are addressed using the full literature ranges of key parameters.

Given the extensive range of optimal temperatures, one can ask how to use the results. It should be noted, however, that dealing with uncertainty and normative choices is part of climate policy decision-making, with or without insights of different models. This will include deciding on acceptable risk levels and normative choices like the discount rate. From our analysis, the risk of high damages appears to be higher than the risk of high mitigation costs. If this is combined with the suggestion of Stern and, more recently, Emmerling et al.[35] that low discount rates are warranted for long-term climate policy, our results confirm that for low discount rates and medium to high damages, cost-optimal temperatures are in line with the long-term objectives of the Paris Agreement. Moreover, the research could possibly reduce some of the uncertainties over time.

## Methods

**The model**. A schematic overview of the model is shown in Fig. 1. The economic module (top grey box) consists of a Cobb-Douglas production function, which calculates GDP using exogenous population and total factor productivity[20]. The GDP is divided between a fixed share to investments and the remaining share to consumption. The investments are added to the global capital stock, which forms together with labour the two production factors for GDP in the next time step. The development of labour is set equal to population developments. The goal is to maximise discounted utility, where utility is an increasing function of consumption.

The next component of the model is the emissions module. $CO_2$ emissions are calculated based on GDP and an emission factor representing the carbon intensity of the energy system (bottom grey box in Fig. 1). The interactions between the emission module and the economic module occur through two mechanisms: damages from climate change, and mitigation costs. Unlike the DICE model, which uses a two-box climate module (which has recently been shown not to be able to reach a 2°C target[36]), the cumulative $CO_2$ emissions in our model are translated into global mean temperature (GMT) through the linear and instantaneous TCRE relation. This simple climate model is shown to provide more realistic outcomes than the DICE climate module[37]. The TCRE includes the effect of non-$CO_2$ emissions, which are therefore implicitly coupled to the $CO_2$ emissions. As suggested in previous research[26], the non-$CO_2$ emissions are correlated with $CO_2$ emissions, making this a reasonable assumption.

The increase in GMT causes GDP loss, quantified through damage functions. To counter these damages, a global carbon price is used at every time period, which causes a reduction of emissions as defined through a quadratic Marginal Abatement Cost (MAC) curve with technological learning (learning-by-doing). The MAC also quantifies the mitigation costs, which are deducted from the GDP, similarly to DICE and FAIR[8,15]. To model the limited availability of negative emission technologies[38], we impose a minimum emission level of −20 $GtCO_2$/yr. This value is based on the minimum emission levels of the scenarios in the scenario explorer for 1.5°C pathways[39] underpinning the IPCC Special Report on Global Warming of 1.5°C[40]. Inertia is modelled by applying a constraint on the difference in $CO_2$ emissions between 2 consecutive years of 2.2 $GtCO_2$/year (based on the maximum reduction speed of the IPCC 1.5°C database[39]). In every experiment, the time horizon is the year 2100, but the optimisation runs throughout the 22nd century to counter end-of-horizon problems.

The optimal carbon price is calculated using the Bellman Equation with as state variables the cumulative emissions and the capital stock, as control variable the carbon price at each time period and as objective function the discounted utility. This methodology is detailed in SI.4.2.

We distinguish between two cases: scenarios with a fixed carbon budget (or temperature target), and scenarios without target. The first case represents a cost-minimising setting while the second constitutes a traditional cost-benefit analysis. In the literature, cost-minimising analysis is typically performed using relatively detailed process models, for instance, to look into the role of specific technologies or determine regional costs. In this approach, it is assumed that climate targets are chosen by policy-makers in international negotiations (based on both monetary and non-monetary information). Cost-benefit models, in contrast, are typically more stylised models that determine an optimal target based on cost-optimisation (which means that all damages need to be expressed in monetary terms). The model here can be used for both types of analysis. It is also able to account for the impact of damages in cost-minimising analysis (which is typically not done).

The full mathematical formulation of the model is available in SI.4.

Although there are some modelling differences between our model and DICE (e.g., different climate module, fixed investment savings rate, endogenous technological change, inclusion of inertia), the main differences reside in the calibration of the parameters.

**Model parameters**. This model allows using literature ranges on parameters relevant for global climate policy. These parameters are highlighted in colour in Fig. 1.

First, the socio-economic variables like population, total factor productivity and baseline emissions intensity are obtained from the SSPs (blue)[20].

The damage functions (green) cover the current literature range by choosing a low, medium and high damage function (see SI.1.3):

- The DICE 2016R2-damage function, representing low damages,
- The medium damage function resulting from a meta-analysis by Howard et al., which they refer to as 'the preferred model for total (non-catastrophic plus catastrophic) damages'. This function is based on empirical damages and traditional estimates like DICE and FUND. We refer to this damage as 'Howard Total'.
- The empirical damage function from Burke, Hsiang and Miguel (2015)[23]. To cover the high end of damages, we use their Long Run (LR) version, which takes into account damages to the GDP growth rate based on the temperature of the 5 previous years. The GDP per capita growth losses are converted to a GDP damage function using an iterative calibration method[13] (see SI.1.3).

The effect of $CO_2$ emissions on global temperature is assumed to be linear and instantaneous through the TCRE relation[37]. Based on the IPCC AR5 Working Group 1[24] relationship, we derive a value of the TCRE between 0.42 and 0.82°C per 1000 GtCO$_2$ (5–95th percentile), with median of 0.62, equal to the range used in van Vuuren et al.[26].

The MAC curve is calibrated to three levels of mitigation costs, using the consumption loss range from the IPCC AR5 Working Group 3, Fig. 6.23[25]. This calibration is performed using quantile regression at the 5th, 50th and 95th percentiles to give a MAC with low, middle and high mitigation costs respectively. More information on the calibration is available in SI.1.

Finally, the utility is discounted at three pure rate of time preference values (also called utility discount rates): the low bound of 0.1%, as used in the Stern review[28], 1.5% and 3%. The latter two values correspond with the values used in DICE-1999 and DICE-2007 (and following versions)[19,41], respectively. These values span a similar range as a recent expert elicitation of social discount rates, where the 5th and 95th percentiles of the PRTP values are 0% and 3.5%/year[12,29], with an average reported value of 1.1%/year.

The effect of using a cubic, instead of a quadratic, MAC is discussed in SI.5.1, as well as the effect of using the full range of PRTP/elasticity of marginal utility combinations from the aforementioned expert elicitation (SI.7).

**Analysing the variance using Sobol decomposition**. Owing to the relative simplicity of this model, we are able to calculate the optimal carbon price path (both with and without carbon budget) for every combination of parameter values shown in Table 1 (405 scenarios). This allows us to analyse the relative importance of each parameter. We quantify the contribution to the variance of each parameter with the Sobol indices[42,43]. These are calculated using a Monte Carlo method. This method requires a distribution for each parameter. However, sampling from a continuous distribution for each parameter would require thousands or millions of runs, which is computationally infeasible. For this reason, we approximate the distribution of each parameter by a discrete distribution best matching the normal distribution of the underlying distribution using only the parameter values available in Table 1. Since the values for the mitigation costs and the TCRE represent the 5th, 50th and 95th percentiles, the discrete distribution with equal mean and variance is a weighted distribution where the median value is 3.4 times more likely to be chosen (see Supplementary Information 4.3). A problem with this method is that the SSP, damage function and discount rate do not have an underlying distribution. To still be able to quantify the relative importance of each parameter, we associate a uniform discrete distribution to these parameters. More details of this method are available in SI.4.3.

## Code availability
The full model code is available at https://github.com/kvanderwijst/DamagesAndCarbonPrice (https://doi.org/10.5281/zenodo.4555423[44]).

## Data availability
The data used for the SSP-related quantities (baseline GDP and population) are available at the IIASA SSP-database: https://tntcat.iiasa.ac.at/SspDb/. The data for each figure and underlying model runs are available at https://github.com/kvanderwijst/DamagesAndCarbonPrice (https://doi.org/10.5281/zenodo.4555423[44]).

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

## Acknowledgements

The research presented in this paper benefitted from funding under the European Union's Horizon 2020 Framework Programme for Research and Innovation under grant agreement no. 776479 for the project CO-designing the Assessment of Climate Change costs. https://www.coacch.eu.

## Author contributions

K.-I.v.d.W., A.H. and D.v.V. contributed to the elaboration of the model, the writing of the article and the design of the experiments. K.-I.v.d.W. developed the model code and analysis.

## Competing interests

The authors declare no competing interests.
