## [Peer Review File · Nature Communications]

REVIEWER COMMENTS

Reviewer #1 (Remarks to the Author):

Integrated assessment models typically either (i) focus on cost effectiveness analysis of a specific mitigation target while including a detailed representation of economic sectors or (ii) calculate an optimal temperature target by the end of century using a cost-benefit approach at the expense of a less detailed representation of the global economy. This paper develops an integrated climate-economy model that is capable of jointly analyzing key parameter uncertainties that have previously been only discussed in one of the two literature strands (and modelling communities) separately. I think the paper addresses an important topic using an appropriate modelling framework. However, I have some concerns that I will summarize in the following:

Major remarks:

- Novelty:

Given that most of the presented sensitivities with respect to the cost-effective and economically optimal emission and carbon price levels have already been analyzed in the existing literature, I am a bit concerned with the presentation of the additional scientific value of the paper. The authors refer to this added value at two points in the paper (lines 98/99 and in line 357) but I think this should be made even clearer in the beginning of the paper.

o In line 357 they write "The added value of our analysis is that it allows to investigate the effect of important, normative assumptions on policy relevant quantities, like the magnitude of the carbon price or the optimal temperature." With respect to the discounting assumptions, for example, the authors mostly follow the old Nordhaus-Stern debate by using only three different estimates of the pure time preference rate as input to the decomposition of uncertainty. Recent publications analyze a much more comprehensive range of the central normative assumptions of social discounting as revealed by expert opinion published in Drupp et. al. 2018. Building on this Hänsel et al. 2020 have investigated how this influences optimal climate policy in an updated version of the DICE integrated assessment model.

o In line 98 the authors write that "this paper presents a comprehensive overview of the relative importance of each of these results" referring to results that have already been analyzed in the literature. I think that bringing together the insights of the two IAM modelling communities (cost-benefit vs. cost-effectiveness analysis) in a transparent meta-model is really the key contribution of the paper and this should be made even clearer. Also it should be made transparent why only a subset of 5 determinants of climate policy (SSP, Damages, TCRE, Mitigation cost level, pure rate of time preference) has been included in the analysis.

- Definition of uncertainty:

In the title the authors claim to provide a "decomposition of uncertainty". The main results of the paper, however, evolve around the values presented in Table 1 and comprise three main estimates per five uncertain parameters analyzed in the paper. Uncertainty, however, is typically defined as a set of future states outcomes with unknown probabilities. The authors present an analysis of the contribution to the variance of each parameters, but this is also based on a Monte Carlo Analysis with known parameter distributions. Hence, I think the authors should be clearer in what the suggested modelling framework can do with respect to analyzing uncertainty. This is, in my view, a joint analysis of the role of different estimates for key sensitivities within the two strands of the literature on IAMs (i.e. cost-effective vs. cost-benefit).

- Similarities with DICE and FAIR:

The authors write that the proposed model "shows some similarities with the DICE and the FAIR model" (lines 67-68). Although the modelling structure is detailed in the supporting information, this should also be made more explicit in the main text. It seems to be that the structure is very similar to a DICE model but uses the FAIR temperature model instead of the standard DICE two-box climate system. Since DICE is considered the most important benchmark for a cost-benefit IAM, the authors should specify in how far the proposed model deviated from DICE and why.

- Presentation of the paper's shortcomings:

Many determinants that drive cost-effective and optimal climate policy have been left out in the analysis. Although it is of course fine to just focus on a subset of drivers, I would find it instructive to include a short discussion of factors that have not been considered in the presented framework. What about the relative scarcity of environmental goods, natural capital in the production function, stochastic uncertainty, endogenous technological change, alternative welfare criteria, inequality and heterogeneity? Just to mention a few.

Minor remarks:

- In figure 2 the y-axis has different scales in panel a-d. Maybe it is more informative to use the same scale to make a comparison of the relative magnitude of each driver possible.

Literature

Drupp, M.A., Freeman, M.C., Groom, B. and F. Nesje (2018). Discounting Disentangled. *American Economic Journal: Economic Policy* 10(4): 109-134.

Hänsel, M.C., Drupp, M.A., Johansson, D.J.H., Nesje, F., Azar, C., Freeman, M.C., Groom, B. and T. Sterner (2020). Climate economics support for the UN climate targets. *Nature Climate Change* 10: 781-789.

Reviewer #2 (Remarks to the Author):

The authors perform a sensitivity analysis of key climate-economic metrics (carbon price, optimal temperature target) to major modelling assumptions within a simple optimization model. They consider both a cost-effective and a cost-benefit setting, and apply Sobol decomposition methods to attribute the observed uncertainties to each studied factor. Since the literature studying economically optimal climate outcomes under uncertainty is not scarce, I think the authors need to put some extra effort on their manuscript before publication.

First, the literature review seems to miss some relevant papers and more in-depth comparisons. For example, [1] builds a probabilistic meta-model from mitigation and damage costs reported in the AR5 database, and derive candidate cost-benefit optimal temperature targets. [2] performs an extensive sampling of the uncertainty of most DICE parameters, and proposes a dynamic sensitivity analysis where climate sensitivity plays a stronger role than suggested here. [3] updates DICE with the latest climate science and economics estimates, which lead to optimal temperature outcomes consistent with the UN proposed targets. Furthermore, results are only critically discussed with respect to Glanemann et al [4], but it would be useful to set the results within the other cited papers as well. As a further example, reference 11, which tries to quantify economically optimal warming, is cited and immediately dismissed as "limited in scope".

Second, more care needs to be taken when claiming that "full literature ranges" are being used. Is it really the case? For example, mitigation costs span a 5-95 percentile range. Does the 0.42-0.82 °C/1000GtCO₂, used for the TCRE, span a similarly extreme range? How does it compare to IPCC climate sensitivity estimates? I would suspect that the choice of this range could drive the relative importance of TCRE in the sensitivity results. Also, while for highly unknown parameters like mitigation costs or SSPs a uniform distribution could be a reasonable approach, for the climate sensitivity (and hence the TCRE) the uniform distribution is not well representative. Does this affect the results? In general, I would make sure that range and distribution choices are well anchored in the literature for the claim above to hold.

Third, some mitigation assumptions deserve further justification. The authors adopt an emission limit of -20GtCO₂/yr without really discussing it. Negative emissions are an important and highly debated assumption, driving both carbon price and temperature outcomes especially for higher discount rates, higher population/GDP trajectories and higher mitigation costs. Some models and recent literature are quite pessimistic on availability of large scale negative emissions. I would

suggest to check whether results hold also when the negative emissions limit is reduced, and to better argue why -20GtCO₂/yr should be the nominal choice. Another two factors that deserve attention are emission reduction speed and shift over time of the MAC (i.e. learning). They are mentioned in the methods and SI but not really discussed in the main text. Also, Figure 2 suggests that emissions in 2020 could be very different depending on the scenario, which might not be consistent with the latest statistics.

Fourth, authors do not discuss interaction effects in their sensitivity analysis. Based on previous studies [5], I would expect that changing SSP and technology costs assumptions together have different effect than changing them in isolation. Also damages and TCRE should be interacting. Figure 6 does show a grey bar for interaction effects, but I would recommend the authors to comment on individual vs interaction vs total effects also in the main text, and given the availability of the full combinatorics of runs, to comment on specific interactions across pairs of factors if of interest for the analysis.

[1] Drouet, L., Bosetti, V., & Tavoni, M. (2015). Selection of climate policies under the uncertainties in the Fifth Assessment Report of the IPCC. *Nature Climate Change*, 5(10), 937–940. <https://doi.org/10.1038/nclimate2721>

[2] Lamontagne, J. R., Reed, P. M., Marangoni, G., Keller, K., & Garner, G. G. (2019). Robust abatement pathways to tolerable climate futures require immediate global action. *Nature Climate Change*, 9(4), 290. <https://doi.org/10.1038/s41558-019-0426-8>

[3] Hänsel, M. C., Drupp, M. A., Johansson, D. J. A., Nesje, F., Azar, C., Freeman, M. C., Groom, B., & Sterner, T. (2020). Climate economics support for the UN climate targets. *Nature Climate Change*, 1–9. <https://doi.org/10.1038/s41558-020-0833-x>

[4] Glanemann, N., Willner, S. N., & Levermann, A. (2020). Paris Climate Agreement passes the cost-benefit test. *Nature Communications*, 11(1), 1–11. <https://doi.org/10.1038/s41467-019-13961-1>

[5] Marangoni, G., Tavoni, M., Bosetti, V., Borgonovo, E., Capros, P., Fricko, O., Gernaat, D. E. H. J., Guivarch, C., Havlik, P., Huppmann, D., Johnson, N., Karkatsoulis, P., Keppo, I., Krey, V., Broin, E. Ó., Price, J., & Vuuren, D. P. van. (2017). Sensitivity of projected long-term CO₂ emissions across the Shared Socioeconomic Pathways. *Nature Climate Change*, 7(2), 113–117. <https://doi.org/10.1038/nclimate3199>

Reviewer #3 (Remarks to the Author):

To the Editor,

I recommend a revise a resubmit for “On the optimality of 2°C targets: a decomposition of uncertainty” by Kaj-Ivar van der Wijst, Andries F. Hof, and Detlef P. van Vuuren. By calibrating their own integrated-assessment model, the authors can make a direct apples-to-apples comparison of marginal abatement costs (MAC) and the social cost of carbon (SCC); cost-minimizing and optimal emission paths; and policy-goal and optimal temperature levels. In doing so, they determine the parameters to which each carbon price is most sensitive. Importantly, they find that for most parameter values, the SCC exceeds the MAC for a 2-degree temperature (as their model frequently finds that the optimal temperature level is below 2 degrees). As the debate between the MAC and SCC has recently heated up (e.g., Kaufman et al. (2020)), I think that this paper makes a key contribution in comparing these two policy instruments and which parameters impact them most in a consistent setting.

Major Suggestions

1. At points, I found this paper difficult to follow as it jumps between socially-optimal and cost-minimizing and between various policy outcomes (price; emissions, and maximum temperature). I think making these transitions clearer would help the paper. One way to reduce confusion is to use different vocabulary for each set of runs. For example, the carbon price equals the marginal abatement cost (MAC) in the cost minimization (i.e., policy evaluation) setting and the social cost of carbon (SCC) in the optimal setting. Similarly, the authors could refer to “cost-minimizing” and

“optimal” emissions paths.

2. For the lower damage function, I would use DICE-2016R2’s damage function. This damage function is lower than DICE-2013R. Normally, I would not make such a suggestion given the nominal difference. However, it represents a more defensible estimate using better data and methods. It also is better documented in Nordhaus and Moffat (2017).

3. The authors cite three damage functions. As Howard and Sterner (2017) provide confidence intervals, it would be helpful to understand whether Burke et al.’s long-run damage function and Nordhaus and Moffat (2017) are contained within the 95th percent confidence interval of Howard and Sterner (2017). If the 95th percent confidence interval of Howard and Sterner (2017) is wider than estimates provided in the literature, then it may make sense to redefine the low and high damage functions in terms of Howard and Sterner (2017)’s 5th and 95th confidence interval.

4. I wonder why the authors do not use Drupp et al. (2018) to determine the central discount rate and their range (or at least compare it to your selected range). For the pure rate of time preference, Drupp et al. (2018) find a mean of 1.1 and median is 0.5. Based on the Heal voting methodology, I believe that 0.5 is the appropriate central estimate (see Howard and Sylvan, 2020). Similarly, the Interagency Working Group on the Social Cost of Carbon (IWG, 2010)’s technical support document specifies a range for the pure rate of time preference.

5. It is not clear why the authors conducted a sensitivity analysis over the elasticity of marginal utility of consumption instead of including it in their analysis like the pure rate of time preference. This is particularly true, as the former parameter interacts with the growth rate of per capita consumption. Again, it can be calibrated by Drupp et al. (2018) or IWG (2010).

6. Given the ongoing debate over the MAC and SCC (e.g., see claims in Kaufman et al. 2020), it would be helpful to measure and discuss the overall uncertainty of the MAC and SCC in addition to their relative values. Similarly, it would be helpful to understand which parameters are the most variable by using something like a coefficient of variation.

7. There is a lot of literature indicating that uncertainty of mitigation costs increases with the aggressiveness of mitigation in a selected emission scenario (for example, Kuik et al., 2009). Is this being captured given the authors’ use of IPCC AR5 consumption losses? Based on the Appendix, the answer appears to be no. While the authors’ argument about canceling impacts make sense, the relative direction of a lower baseline and fewer technologies available could matter as acknowledged by the authors when they state the effects only partially cancel out. At a minimum, the impact of assuming away this issue should be discussed.

Minor Suggestions

1. In your citations, I would include Hänsel et al. (2020)

2. This is a particularly confusing sentence: “Consequently, choices on discount rate and importance of damages have to be made now to reduce the uncertainty in climate policy: these choices have a strong effect on the cost-effective price of today.”

3. You should contrast your results with DICE-2016R2, which finds that a 2-degree limit is impossible to reach due to its climate dynamics (i.e., calibration); see Hänsel et al. (2020) and Howard and Sylvan (2020).

4. When the authors first refer to damage costs, I think that clarifying that they are considering damages as a cost here would make the paper clearer.

5. In their sensitivity analysis, the authors explore the sensitivity of their results to the MAC curve. Similarly, I think that they should explore the shape of the damage function. This is possible using Howard and Sterner (2017)’s appendix. An additional sensitivity analysis that may be of interest is the impact of turning off learning by doing given the concerns raised by Nordhaus (2014).

6. Also, Howard and Sterner (2017)'s preferred specification is the non-catastrophic damage function. The total damage function includes catastrophic impacts, which may proxy for tipping points. Is there a reason that the authors selected total instead of non-catastrophic? Please explain. If it is included as a proxy for the costs of tipping points, this should be mentioned when tipping points are discussed in lines 344 to 347.

References

Hänsel, M. C., Drupp, M. A., Johansson, D. J., Nesje, F., Azar, C., Freeman, M. C., ... & Sterner, T. (2020). Climate economics support for the UN climate targets. *Nature Climate Change*, 10(8), 781-789.

Howard, P. H., & Sylvan, D. (2020). Wisdom of the experts: Using survey responses to address positive and normative uncertainties in climate-economic models. *Climatic Change*, 1-20.

Kaufman, N., Barron, A. R., Krawczyk, W., Marsters, P., & McJeon, H. (2020). A near-term to net zero alternative to the social cost of carbon for setting carbon prices. *Nature Climate Change*, 1-5.

Kuik, O., Brander, L., & Tol, R. S. (2009). Marginal abatement costs of greenhouse gas emissions: A meta-analysis. *Energy policy*, 37(4), 1395-1403.

Nordhaus, W. D. (2014). The perils of the learning model for modeling endogenous technological change. *The Energy Journal*, 35(1).

Nordhaus, W. D., & Moffat, A. (2017). A survey of global impacts of climate change: replication, survey methods, and a statistical analysis (No. w23646). National Bureau of Economic Research.

REVIEWER COMMENTS

Reviewer #1 (Remarks to the Author):

Integrated assessment models typically either (i) focus on cost effectiveness analysis of a specific mitigation target while including a detailed representation of economic sectors or (ii) calculate an optimal temperature target by the end of century using a cost-benefit approach at the expense of a less detailed representation of the global economy. This paper develops an integrated climate-economy model that is capable of jointly analyzing key parameter uncertainties that have previously been only discussed in one of the two literature strands (and modelling communities) separately. I think the paper addresses an important topic using an appropriate modelling framework. However, I have some concerns that I will summarize in the following:

Thank you very much for your review and your comments. We have addressed your remarks point by point below.

Major remarks:

- Novelty:

Given that most of the presented sensitivities with respect to the cost-effective and economically optimal emission and carbon price levels have already been analyzed in the existing literature, I am a bit concerned with the presentation of the additional scientific value of the paper. The authors refer to this added value at two points in the paper (lines 98/99 and in line 357) but I think this should be made even clearer in the beginning of the paper.

The added value of our paper is that we analyse the impact of full uncertainty ranges of the most important determinants on cost-minimising and cost-benefit studies. This has, to our knowledge, not yet done before. We have therefore adapted the abstract and introduction to make our added value clearer. More specifically, we have added in the abstract that the meta-model is able to disentangle the different uncertainties using full literature ranges and is used to directly compare the insights of the cost-minimising and cost-benefit modelling communities.

o In line 357 they write “The added value of our analysis is that it allows to investigate the effect of important, normative assumptions on policy relevant quantities, like the magnitude of the carbon price or the optimal temperature.” With respect to the discounting assumptions, for example, the authors mostly follow the old Nordhaus-Stern debate by using only three different estimates of the pure time preference rate as input to the decomposition of uncertainty. Recent publications analyze a much more comprehensive range of the central normative assumptions of social discounting as revealed by expert opinion published in Drupp et. al. 2018. Building on this Hänsel et al. 2020 have investigated how this influences optimal climate policy in an updated version of the DICE integrated assessment model.

Thank you for pointing us to these references, which we have now added to our manuscript. In Drupp et al (2018), the 5th and 95th percentile of the responses for the pure rate of time preference are 0% and 3.5%/year. We believe that this provides a good extra basis for our chosen range, spanning from 0.1% to 3%. Our values have the advantage of being values widely used in past literature.

However, the Drupp et al expert elicitation also provides values for the elasticity of marginal utility (elasmu) corresponding to each PRTP values. The 5th, 50th and 95th percentiles of the social discount

rate (combining PRTP and elasmu to one value) give a wider range of discounting than the values we use in our main analysis (see the table in SI.7).

This wider range does not significantly affect our results, since the Drupp et al values have an underlying distribution (which is roughly normal, as shown in Supplementary Figure 7.1). Therefore, the outer values (5 and 95th perc.) should be given less weight than the median value (in contrast to the uniform distribution we use for our PRTP values).

We have repeated our whole analysis using the distribution of Drupp et al in SI 7 (and discussed this in the Discussion of the main text). This confirms the suggestion that the difference with our three PRTP values is not so large.

o In line 98 the authors write that “this paper presents a comprehensive overview of the relative importance of each of these results” referring to results that have already been analyzed in the literature. I think that bringing together the insights of the two IAM modelling communities (cost-benefit vs. cost-effectiveness analysis) in a transparent meta-model is really the key contribution of the paper and this should be made even clearer. Also it should be made transparent why only a subset of 5 determinants of climate policy (SSP, Damages, TCRE, Mitigation cost level, pure rate of time preference) has been included in the analysis.

In line with your first suggestion, we have included the key contribution both in the abstract and introduction. In the introduction and more predominantly in the discussion, we have explained why we focus on these 5 determinants of climate policy. Most importantly, existing literature points out that these 5 uncertainties have the largest impact on optimal temperature outcomes (also see our response on “Presentation of the paper’s shortcomings” below).

- Definition of uncertainty:

In the title the authors claim to provide a “decomposition of uncertainty”. The main results of the paper, however, evolve around the values presented in Table 1 and comprise three main estimates per five uncertain parameters analyzed in the paper. Uncertainty, however, is typically defined as a set of future states outcomes with unknown probabilities. The authors present an analysis of the contribution to the variance of each parameters, but this is also based on a Monte Carlo Analysis with known parameter distributions. Hence, I think the authors should be clearer in what the suggested modelling framework can do with respect to analyzing uncertainty. This is, in my view, a joint analysis of the role of different estimates for key sensitivities within the two strands of the literature on IAMs (i.e. cost-effective vs. cost-benefit).

Thank you for this suggestion. We have included a new section in the discussion on what we understand with uncertainty – and what the added value would be of including stochastic uncertainty. This would mainly be the inclusion of climate and socio-economic tipping points.

- Similarities with DICE and FAIR:

The authors write that the proposed model “shows some similarities with the DICE and the FAIR model” (lines 67-68). Although the modelling structure is detailed in the supporting information, this should also be made more explicit in the main text. It seems to be that the structure is very similar to a DICE model but uses the FAIR temperature model instead of the standard DICE two-box climate system. Since DICE is considered the most important benchmark for a cost-benefit IAM, the authors should specify in how far the proposed model deviated from DICE and why.

We have made this more explicit in the Method section. Indeed, the main modelling difference is the different climate module. The largest differences however reside in the calibration of parameters (calibration based on SSPs and on full literature range).

- Presentation of the paper's shortcomings:

Many determinants that drive cost-effective and optimal climate policy have been left out in the analysis. Although it is of course fine to just focus on a subset of drivers, I would find it instructive to include a short discussion of factors that have not been considered in the presented framework. What about the relative scarcity of environmental goods, natural capital in the production function, stochastic uncertainty, endogenous technological change, alternative welfare criteria, inequality and heterogeneity? Just to mention a few.

We have extended the discussion on sensitivity runs.

- Scarcity of environmental goods, natural capital in the production function: the recent work of Bastien-Olvera and Moore has provided very interesting quantifications of this. However we believe that this goes beyond the scope of our paper, where we investigate the literature ranges of the more traditional uncertain parameters. Moreover, since valuing biodiversity has always been an uncertain part in creating damage functions, we argue that we take this into account by using a large range of damage functions.
- Stochastic uncertainty: see above on the point on "uncertainty".
- Endogenous technological change: different assumptions on the magnitude of technological learning are already included in the range of mitigation costs. Explicitly including different values of learning by doing would only slightly change the shape of the carbon price path, and bring mitigation effort towards the present. However, since the effect of different learning by doing values only is rather small (we are currently publishing a study on this), we have not included it in our analysis. We have mentioned it in the discussion though.
- Alternative welfare: we have moved away from a simple discussion on elasmu 1.001 vs 1.45, and included the welfare specification (PRTP/elasmu pairs) of Drupp et al in the discussion and in SI.7. However, other specifications could also be investigated, like moving away from the Ramsey rule (as mentioned in Drupp et al). This is closely related to your point on inequality and heterogeneity. While this is very interesting, this cannot be calculated with a global (meta)model. We have also extended the discussion with this.

Minor remarks:

- In figure 2 the y-axis has different scales in panel a-d. Maybe it is more informative to use the same scale to make a comparison of the relative magnitude of each driver possible.

This has now been fixed.

Literature

Drupp, M.A., Freeman, M.C., Groom, B. and F. Nesje (2018). Discounting Disentangled. American Economic Journal: Economic Policy 10(4): 109-134.

Hänsel, M.C., Drupp, M.A., Johansson, D.J.H., Nesje, F., Azar, C., Freeman, M.C., Groom, B. and T. Sterner (2020). Climate economics support for the UN climate targets. *Nature Climate Change* 10: 781–789.

Reviewer #2 (Remarks to the Author):

The authors perform a sensitivity analysis of key climate-economic metrics (carbon price, optimal temperature target) to major modelling assumptions within a simple optimization model. They consider both a cost-effective and a cost-benefit setting, and apply Sobol decomposition methods to attribute the observed uncertainties to each studied factor. Since the literature studying economically optimal climate outcomes under uncertainty is not scarce, I think the authors need to put some extra effort on their manuscript before publication.

Thank you for your review.

First, the literature review seems to miss some relevant papers and more in-depth comparisons. For example, [1] builds a probabilistic meta-model from mitigation and damage costs reported in the AR5 database, and derive candidate cost-benefit optimal temperature targets. [2] performs an extensive sampling of the uncertainty of most DICE parameters, and proposes a dynamic sensitivity analysis where climate sensitivity plays a stronger role than suggested here. [3] updates DICE with the latest climate science and economics estimates, which lead to optimal temperature outcomes consistent with the UN proposed targets. Furthermore, results are only critically discussed with respect to Glanemann et al [4], but it would be useful to set the results within the other cited papers as well. As a further example, reference 11, which tries to quantify economically optimal warming, is cited and immediately dismissed as "limited in scope".

Thank you for these extra references. We have now discussed them in the introduction and, where possible, made a more direct comparison with their results in the results section. Specifically, here are the differences per paper:

[1] (Drouet et al, 2015): While Drouet et al use more detailed emission pathways generated using detailed process IAMs, they were all calculated without taking damages into account. We go beyond the selection of a carbon budget according to specific rules, and rather calculate the optimal emission pathway endogenously, where each of the variables are weighted throughout the whole time period. As for the results, the optimal carbon budget candidates selected by Drouet et al are quite significantly higher than our optimal temperatures/carbon budgets, mainly due to the much lower damage functions employed in their paper, as shown in Supplementary Figure S7 of [1]. We have chosen not to discuss this paper in more detail in the Discussion paragraph about recent literature since its damage functions have become outdated (which is now mentioned in the introduction).

[2] (Lamontagne et al, 2019): The main difference with our paper is that Lamontagne et al consider the abatement growth rate (and therefore the whole emission pathway) as an uncertainty parameter. This means that their model is run in simulation mode, whereas we optimise. Moreover, both their damage parameter range (combination of parameters a1, a2 and a3) and their mitigation cost range (combination of exponent of the MAC and cost of backstop) are much smaller than our

ranges. They cover the regression uncertainty of these parameters, but definitely do not cover the literature ranges of the damage and mitigation costs. The uncertainty in their climate model parameters do capture the climate model uncertainty. As a result, the impact of the uncertainty in the climate model is much larger in their paper than in our results.

[3] (Hänsel et al, 2020) Hänsel et al focus on cost-benefit analysis using the full range of social discounting parameter values, like pure rate of time preference and elasticity of marginal utility. Our results are similar to Hänsel et al's results, if we use the Howard's damage function and median estimates of mitigation costs, and the full range of PRTP values. However, we consider a much larger range of values for the parameters other than discounting. This leads to a wider range of optimal end of century temperatures.

Second, more care needs to be taken when claiming that "full literature ranges" are being used. Is it really the case? For example, mitigation costs span a 5-95 percentile range. Does the 0.42-0.82 °C/1000GtCO₂, used for the TCRE, span a similarly extreme range? How does it compare to IPCC climate sensitivity estimates? I would suspect that the choice of this range could drive the relative importance of TCRE in the sensitivity results. Also, while for highly unknown parameters like mitigation costs or SSPs a uniform distribution could be a reasonable approach, for the climate sensitivity (and hence the TCRE) the uniform distribution is not well representative. Does this affect the results? In general, I would make sure that range and distribution choices are well anchored in the literature for the claim above to hold.

Indeed, both the mitigation cost calibration and the values of the TCRE span the 5-95th percentile range. More details on this calculation are available in van Vuuren (2020) (<https://doi.org/10.1038/s41558-020-0732-1>). We have made this clearer in the main text. Since these values are the 5th, 50th and 95th percentiles from a normal distribution, the discrete approximation of this distribution can indeed not be a uniform discrete distribution. Indeed (as detailed in SI 4.3), we used a weighted discrete distribution where we give 3.4 times more weight to the median value as to the 5th and 95th percentile values. We have also included this in the Methods section now. The other parameters (SSP, discount rate and climate damage) use a uniform discrete distribution.

Third, some mitigation assumptions deserve further justification. The authors adopt an emission limit of -20GtCO₂/yr without really discussing it. Negative emissions are an important and highly debated assumption, driving both carbon price and temperature outcomes especially for higher discount rates, higher population/GDP trajectories and higher mitigation costs. Some models and recent literature are quite pessimistic on availability of large scale negative emissions. I would suggest to check whether results hold also when the negative emissions limit is reduced, and to better argue why -20GtCO₂/yr should be the nominal choice. Another two factors that deserve attention are emission reduction speed and shift over time of the MAC (i.e. learning). They are mentioned in the methods and SI but not really discussed in the main text. Also, Figure 2 suggests that emissions in 2020 could be very different depending on the scenario, which might not be consistent with the latest statistics.

Negative emissions: The maximum value of negative emissions is based on the minimum emission levels of the scenarios in the scenario explorer for 1.5°C pathways underpinning the IPCC Special Report on Global Warming of 1.5°C (<https://data.ene.iiasa.ac.at/iamc-1.5c-explorer>); we have added this in the paper. As indeed some critique on negative emissions have been given – we tested the

impact of excluding negative emissions on the results in the Appendix (and have added this in the Discussion).

Learning: we have included a more detailed description of endogenous technological learning, and the impact of changing the learning parameter, in the discussion.

Different emissions in 2020: in our original manuscript, we ran the model by default without inertia constraints, leading to immediate abatement in 2020. While this has only a very slight influence on long-term climate goals (except for low costs/low discounting scenarios), we agree that the results in Figure 2 are confusing. For this reason, we have switched to the inertia constraint by default (the only downside of this was the extra computing time since an extra dimension to the optimisation state variable needed to be added).

Fourth, authors do not discuss interaction effects in their sensitivity analysis. Based on previous studies [5], I would expect that changing SSP and technology costs assumptions together have different effect than changing them in isolation. Also damages and TCRE should be interacting. Figure 6 does show a grey bar for interaction effects, but I would recommend the authors to comment on individual vs interaction vs total effects also in the main text, and given the availability of the full combinatorics of runs, to comment on specific interactions across pairs of factors if of interest for the analysis.

Splitting up the Sobol decomposition into higher order interaction terms is indeed interesting. We have added this decomposition in Supplementary Figure 3.8, and have discussed it in the main text. The largest interaction terms are SSP/Damage and TCRE/Damage. This makes sense since the damage function makes up most of the variation in optimal temperature. Moreover, the largest interaction term which does not include damages is indeed the SSP/Mitigation cost term, partly due to the high influence of GDP per capita and technology improvement on the variance within SSPs as shown by [5].

[1] Drouet, L., Bosetti, V., & Tavoni, M. (2015). Selection of climate policies under the uncertainties in the Fifth Assessment Report of the IPCC. *Nature Climate Change*, 5(10), 937–940.

<https://doi.org/10.1038/nclimate2721>

[2] Lamontagne, J. R., Reed, P. M., Marangoni, G., Keller, K., & Garner, G. G. (2019). Robust abatement pathways to tolerable climate futures require immediate global action. *Nature Climate Change*, 9(4), 290. <https://doi.org/10.1038/s41558-019-0426-8>

[3] Hänsel, M. C., Drupp, M. A., Johansson, D. J. A., Nesje, F., Azar, C., Freeman, M. C., Groom, B., & Sterner, T. (2020). Climate economics support for the UN climate targets. *Nature Climate Change*, 1–9. <https://doi.org/10.1038/s41558-020-0833-x>

[4] Glanemann, N., Willner, S. N., & Levermann, A. (2020). Paris Climate Agreement passes the cost-benefit test. *Nature Communications*, 11(1), 1–11. <https://doi.org/10.1038/s41467-019-13961-1>

[5] Marangoni, G., Tavoni, M., Bosetti, V., Borgonovo, E., Capros, P., Fricko, O., Gernaat, D. E. H. J., Guivarch, C., Havlik, P., Huppmann, D., Johnson, N., Karkatsoulis, P., Keppo, I., Krey, V., Broin, E. Ó., Price, J., & Vuuren, D. P. van. (2017). Sensitivity of projected long-term CO₂ emissions across the Shared Socioeconomic Pathways. *Nature Climate Change*, 7(2), 113–117.

<https://doi.org/10.1038/nclimate3199>

Reviewer #3 (Remarks to the Author):

To the Editor,

I recommend a revise a resubmit for “On the optimality of 2°C targets: a decomposition of uncertainty” by Kaj-Ivar van der Wijst, Andries F. Hof, and Detlef P. van Vuuren. By calibrating their own integrated-assessment model, the authors can make a direct apples-to-apples comparison of marginal abatement costs (MAC) and the social cost of carbon (SCC); cost-minimizing and optimal emission paths; and policy-goal and optimal temperature levels. In doing so, they determine the parameters to which each carbon price is most sensitive. Importantly, they find that for most parameter values, the SCC exceeds the MAC for a 2-degree temperature (as their model frequently finds that the optimal temperature level is below 2 degrees). As the debate between the MAC and SCC has recently heated up (e.g., Kaufman et al. (2020)), I think that this paper makes a key contribution in comparing these two policy instruments and which parameters impact them most in a consistent setting.

Thank you very much for your review and valuable comments.

Major Suggestions

1. At points, I found this paper difficult to follow as it jumps between socially-optimal and cost-minimizing and between various policy outcomes (price; emissions, and maximum temperature). I think making these transitions clearer would help the paper. One way to reduce confusion is to use different vocabulary for each set of runs. For example, the carbon price equals the marginal abatement cost (MAC) in the cost minimization (i.e., policy evaluation) setting and the social cost of carbon (SCC) in the optimal setting. Similarly, the authors could refer to “cost-minimizing” and “optimal” emissions paths.

Thank you. Discussing the results of cost-benefit (socially-optimal) and cost-minimising paradigms in a single paper (and based on a single model) is one of the strengths of the paper. It is indeed important that the reader is able to follow what we are doing and so we emphasised the transitions in the paper better and improved the captions of the figures. We also moved away from cost-effective to cost-minimising as per your suggestion (which we find indeed clearer). We do avoid the word SCC given that there is currently still some confusion in literature on its exact definition.

2. For the lower damage function, I would use DICE-2016R2’s damage function. This damage function is lower than DICE-2013R. Normally, I would not make such a suggestion given the nominal difference. However, it represents a more defensible estimate using better data and methods. It also is better documented in Nordhaus and Moffat (2017).

Good point, we have updated the DICE damage function to the DICE-2016R2 specification. This leads to slightly higher optimal 2100 temperatures for the DICE damage function.

3. The authors cite three damage functions. As Howard and Sterner (2017) provide confidence intervals, it would be helpful to understand whether Burke et al.’s long-run damage function and Nordhaus and Moffat (2017) are contained within the 95th percent confidence interval of Howard and Sterner (2017). If the 95th percent confidence interval of Howard and Sterner (2017) is wider than estimates provided in the literature, then it may make sense to redefine the low and high damage functions in terms of Howard and Sterner (2017)’s 5th and 95th confidence interval.

We have now included the 5th and 95th percentile range of the damage function as provided by Howard and Sterner (2017) in Supplementary Figure 1.4. The range of damage functions used in our analysis is in fact larger than this 5-95th range.

4. I wonder why the authors do not use Drupp et al. (2018) to determine the central discount rate and their range (or at least compare it to your selected range). For the pure rate of time preference, Drupp et al. (2018) find a mean of 1.1 and median is 0.5. Based on the Heal voting methodology, I believe that 0.5 is the appropriate central estimate (see Howard and Sylvan, 2020). Similarly, the Interagency Working Group on the Social Cost of Carbon (IWG, 2010)'s technical support document specifies a range for the pure rate of time preference.

Thank you for pointing us to Drupp et al, which we have now added to our manuscript. In Drupp et al (2018), the 5th and 95th percentile of the responses for the pure rate of time preference are 0% and 3.5%/year. We believe that this provides a good extra basis for the range used already in the article, spanning from 0.1% to 3% and which were based on existing literature. Other elements provided by Drupp, however, are the distribution and the uncertainty assigned to elasmu. As clearly discount rates represent a choice, in the main manuscript, we used our original uniform distribution – but in the appendix (SI 7) we used the full Drupp et al setting and redid all analysis for comparison. Interestingly the impact on the contribution to uncertainty are limited (see our response to reviewer #1).

5. It is not clear why the authors conducted a sensitivity analysis over the elasticity of marginal utility of consumption instead of including it in their analysis like the pure rate of time preference. This is particularly true, as the former parameter interacts with the growth rate of per capita consumption. Again, it can be calibrated by Drupp et al. (2018) or IWG (2010).

As an addition to the previous review point and according to this point, we have included the Drupp et al expert elicitation values as alternative welfare specification in the discussion and in SI.7. While looking at the range of elasmu of Drupp et al individually, this would lead to a strong contribution to the uncertainty of elasmu (since it spans values from 0 all the way to 5). However, they are strongly related to the PRTP values. Coupling them and calculating the corresponding social discount rate allows us to obtain 5-50-95th percentiles of the PRTP/elasmu pairs. Since we now can use the underlying distribution (approximately normal, as shown in Supplementary Figure 7.1), we can give less weight to the 5 and 95th p values, just like we do with the cost level and TCRC. All in all, the impact of this new specification of discount uncertainty is small (SI 7).

6. Given the ongoing debate over the MAC and SCC (e.g., see claims in Kaufman et al. 2020), it would be helpful to measure and discuss the overall uncertainty of the MAC and SCC in addition to their relative values. Similarly, it would be helpful to understand which parameters are the most variable by using something like a coefficient of variation.

We can confirm the claims by Kaufman et al (2020) that the social cost of carbon is very uncertain. We have added the variance decomposition of the carbon price in cost-benefit setting (Suppl Figure 3.4b) – in addition to Suppl Fig 3.4a which already showed the relative carbon price uncertainty decomposition. This is also mentioned now in the main text under “Shape of optimal carbon price paths in cost-benefit setting”. The main differences between MAC and SCC uncertainty are 1) the linear increase in uncertainty over time for SCC (in contrast to the different shape of cost-minimising carbon prices, reflecting the different carbon price trajectories) and 2) the smaller influence of mitigation cost level and higher influence of damages in cost-benefit setting versus cost-minimising

setting.

7. There is a lot of literature indicating that uncertainty of mitigation costs increases with the aggressiveness of mitigation in a selected emission scenario (for example, Kuik et al., 2009). Is this being captured given the authors' use of IPCC AR5 consumption losses? Based on the Appendix, the answer appears to be no. While the authors' argument about canceling impacts make sense, the relative direction of a lower baseline and fewer technologies available could matter as acknowledged by the authors when they state the effects only partially cancel out. At a minimum, the impact of assuming away this issue should be discussed.

Our mitigation cost curves do indeed diverge for lower carbon budgets, in line with Kuik et al (2009) (Supplementary Figure 1.2 in our appendix). As we focus on the relevance of the Paris Agreement, we propose not to include more carbon budgets given the already extensive scope of the paper. However, we have done this analysis recently in van Vuuren et al (2020), where the increasing mitigation cost uncertainty for more stringent carbon budgets was indeed confirmed. We used the effect of cancelling impacts mostly to motivate why we use the same MAC for each SSP. We do however calibrate the MAC to the 5th, 50th and 95th percentiles of the IPCC AR5 consumption losses to capture the increasing cost uncertainty for more stringent carbon budgets.

Minor Suggestions

1. In your citations, I would include Hänsel et al. (2020)

Indeed, we have included this paper now.

2. This is a particularly confusing sentence: "Consequently, choices on discount rate and importance of damages have to be made now to reduce the uncertainty in climate policy: these choices have a strong effect on the cost-effective price of today."

We have rephrased this sentence.

3. You should contrast your results with DICE-2016R2, which finds that a 2-degree limit is impossible to reach due to its climate dynamics (i.e., calibration); see Hänsel et al. (2020) and Howard and Sylvan (2020).

Our results show that with the DICE-2016R2 damage function, a 2-degree temperature in 2100 is only optimal when using a low PRTP and assuming low mitigation costs. In all other combinations, the optimal temperature is well above 2C. However, since we use a different climate module than DICE-2016R2, the 2-degree goal is not impossible.

4. When the authors first refer to damage costs, I think that clarifying that they are considering damages as a cost here would make the paper clearer.

Yes, this is now improved.

5. In their sensitivity analysis, the authors explore the sensitivity of their results to the MAC curve. Similarly, I think that they should explore the shape of the damage function. This is possible using Howard and Sterner (2017)'s appendix. An additional sensitivity analysis that may be of interest is the impact of turning off learning by doing given the concerns raised by Nordhaus (2014).

Different shape of the damage function: we have now included the 5-95th percentile range of Howard and Sterner (2017). However, since we already span a large range of damage functions by using the very low (DICE 2016) and very high (Burke Long Run) damage specification, and since the different shape of the damage function mostly influences damages at the high temperature ranges (which are not reached in either cost-effectiveness or cost-benefit), we believe we have covered a sufficiently large range of damage functions.

We have also included more discussion on the learning by doing parameter. As mentioned now in the text, the inclusion of learning by doing mostly shifts mitigation effort towards the future, and has an impact (although limited impact) on the shape of the carbon path trajectory. Of course, more or less technological learning will decrease or increase the overall mitigation costs – however, these differences are captured in the mitigation cost parameter.

6. Also, Howard and Sterner (2017)'s preferred specification is the non-catastrophic damage function. The total damage function includes catastrophic impacts, which may proxy for tipping points. Is there a reason that the authors selected total instead of non-catastrophic? Please explain. If it is included as a proxy for the costs of tipping points, this should be mentioned when tipping points are discussed in lines 344 to 347.

Indeed, our choice was based on both the fact that the total damage function includes catastrophic impacts which are otherwise difficult to damage (without using a full-on stochastic model, and by the way even then it is still very unclear how to quantify these tipping points), and the fact that it provided a good middle damage function between the low (DICE) and high (Burke LR) damage function.

References

Hänsel, M. C., Drupp, M. A., Johansson, D. J., Nesje, F., Azar, C., Freeman, M. C., ... & Sterner, T. (2020). Climate economics support for the UN climate targets. *Nature Climate Change*, 10(8), 781-789.

Howard, P. H., & Sylvan, D. (2020). Wisdom of the experts: Using survey responses to address positive and normative uncertainties in climate-economic models. *Climatic Change*, 1-20.

Kaufman, N., Barron, A. R., Krawczyk, W., Marsters, P., & McJeon, H. (2020). A near-term to net zero alternative to the social cost of carbon for setting carbon prices. *Nature Climate Change*, 1-5.

Kuik, O., Brander, L., & Tol, R. S. (2009). Marginal abatement costs of greenhouse gas emissions: A meta-analysis. *Energy policy*, 37(4), 1395-1403.

Nordhaus, W. D. (2014). The perils of the learning model for modeling endogenous technological change. *The Energy Journal*, 35(1).

Nordhaus, W. D., & Moffat, A. (2017). A survey of global impacts of climate change: replication, survey methods, and a statistical analysis (No. w23646). National Bureau of Economic Research.

REVIEWERS' COMMENTS

Reviewer #1 (Remarks to the Author):

The authors have now addressed my concerns appropriately and thus I recommend publication of the manuscript.

Reviewer #2 (Remarks to the Author):

While the authors did take steps to address part of my points, major concerns remain.

On the literature review, it does not seem the authors have done any meaningful revision. It is curious that they partially address this point in the responses but not in the paper itself. In the main text, they quickly dismiss Drouet et al. [1], which I think is quite relevant, because of outdated damage functions. Yet the authors still consider a similarly wide range of damage functions (from low to catastrophic), and assign the same probability to older and newer formulations, as learning in the climate impact fields is itself very hard. Meanwhile Drouet et al. consider 750 mitigation trajectories derived from process-based IAMs, 38 climate models and 3 damage function shapes, versus the 405 scenarios run with a relatively simple meta-model. Lamontagne et al. [2] might be narrow in mitigation costs uncertainty ranges, but its sensitivity analysis includes almost all DICE parameters and looks at millions of samples, versus the 3-level sampling across 4 dimensions times 5 SSPs of this work. Again, the manuscript lacks any constructive comparison.

This leads to the major remaining problem of this paper: it is hard to appreciate its novelty. Readers still do not know how the authors' results stand against the literature, which already contains similar and under some aspects more sophisticated analyses as shown above. The problem is even deeper once the authors mention van Vuuren et al. [3] in their responses. This paper, which I somehow missed in the first round, shares the same mathematical model of this work, and does discuss sources of uncertainties for different levels of warming. Yet it is only cited for methodological considerations here: no critical evaluation of the advancement of this paper over [3] is done. If the papers are too close, novelty is compromised.

When responding to my concerns on fixed technical change and negative emissions assumptions for such a sensitivity, the authors still provided little evidence to their claims. They write that "Endogenous technological growth is kept fixed at the medium value of van Vuuren [...] Using the full range here would only slightly impact the shape of the carbon price path". It is not clear whether the last statement is supported by any analysis or reference." They also write that "changing the minimum emission level from -20GtCO₂ to 0 (therefore avoiding an emission/temperature overshoot) only has a small effect on the result", with some further reporting, but without properly explaining such surprising finding.

[1] Drouet, L., Bosetti, V., & Tavoni, M. (2015). Selection of climate policies under the uncertainties in the Fifth Assessment Report of the IPCC. *Nature Climate Change*, 5(10), 937–940. <https://doi.org/10.1038/nclimate2721>

[2] Lamontagne, J. R., Reed, P. M., Marangoni, G., Keller, K., & Garner, G. G. (2019). Robust abatement pathways to tolerable climate futures require immediate global action. *Nature Climate Change*, 9(4), 290. <https://doi.org/10.1038/s41558-019-0426-8>

[3] van Vuuren, D. P., van der Wijst, K. I., Marsman, S., van den Berg, M., Hof, A. F., & Jones, C. D. (2020). The costs of achieving climate targets and the sources of uncertainty. *Nature Climate Change*, 10(4), 329–334.

Reviewer #3 (Remarks to the Author):

I will keep this review short, as I have previously reviewed this paper.

I recommend an acceptance of "On the optimality of 2°C targets: a decomposition of uncertainty" by Kaj-Ivar van der Wijst, Andries F. Hof, and Detlef P. van Vuuren. The authors have done a good job of addressing my previous comments. Where they did not make my suggested changes, I believe that their explanations were thoughtful and sufficient.

My only minor suggestion is to rewrite the sentence "A second category consists of models that determine optimal pathways which minimise the costs and benefits of climate policy." Specifically, I think that the sentence should refer to the "maximization of net benefits" or the "minimization of net costs." I prefer the former as it is more standard, though the latter is also acceptable.

REVIEWERS' COMMENTS

Reviewer #1 (Remarks to the Author):

The authors have now addressed my concerns appropriately and thus I recommend publication of the manuscript.

Thank you.

Reviewer #2 (Remarks to the Author):

While the authors did take steps to address part of my points, major concerns remain.

Thank you for your extra comments.

On the literature review, it does not seem the authors have done any meaningful revision. It is curious that they partially address this point in the responses but not in the paper itself. In the main text, they quickly dismiss Drouet et al. [1], which I think is quite relevant, because of outdated damage functions. Yet the authors still consider a similarly wide range of damage functions (from low to catastrophic), and assign the same probability to older and newer formulations, as learning in the climate impact fields is itself very hard. Meanwhile Drouet et al. consider 750 mitigation trajectories derived from process-based IAMs, 38 climate models and 3 damage function shapes, versus the 405 scenarios run with a relatively simple meta-model. Lamontagne et al. [2] might be narrow in mitigation costs uncertainty ranges, but its sensitivity analysis includes almost all DICE parameters and looks at millions of samples, versus the 3-level sampling across 4 dimensions times 5 SSPs of this work. Again, the manuscript lacks any constructive comparison.

We have now included a short discussion of Drouet et al in the Discussion, with the same comments we already included in our previous Response to Reviewer. Concerning the second reference (Lamontagne et al), we already included a more in depth comparison in the Discussion, which the reviewer might have missed.

This leads to the major remaining problem of this paper: it is hard to appreciate its novelty. Readers still do not know how the authors' results stand against the literature, which already contains similar and under some aspects more sophisticated analyses as shown above. The problem is even deeper once the authors mention van Vuuren et al. [3] in their responses. This paper, which I somehow missed in the first round, shares the same mathematical model of this work, and does discuss sources of uncertainties for different levels of warming. Yet it is only cited for methodological considerations here: no critical evaluation of the advancement of this paper over [3] is done. If the papers are too close, novelty is compromised.

While the editor already dismissed the novelty concerns, we would like to stress that van Vuuren et al only focuses on decomposing the uncertainty in mitigation costs – it does not use an IAM of any kind and does not perform any optimisations, nor does it consider damages. Indeed, only the Sobol analysis method is shared, as well as the calibration of mitigation cost uncertainty.

When responding to my concerns on fixed technical change and negative emissions assumptions for such a sensitivity, the authors still provided little evidence to their claims. They write that

"Endogenous technological growth is kept fixed at the medium value of van Vuuren [...] Using the full range here would only slightly impact the shape of the carbon price path". It is not clear whether the last statement is supported by any analysis or reference." They also write that "changing the minimum emission level from -20GtCO₂ to 0 (therefore avoiding an emission/temperature overshoot) only has a small effect on the result", with some further reporting, but without properly explaining such surprising finding.

- Endogenous technological growth: changing the learning rate means that the MAC needs to be recalibrated accordingly. Otherwise, a higher learning rate would lead to costs lower than the calibrated IPCC AR5 range, and vice versa. These two factors (a changed learning rate and a changed MAC calibration) counterbalance each other. Previous research (like the iconic Goulder and Mathai paper from 2000) has also shown that learning by doing has only a very small effect on the carbon price (given a proper calibration). Because of this extra complexity and possible confusion, and limited effect, we have decided to remove the sentence from the discussion. Moreover, we are working on a paper which investigates exactly this issue (which is now in second stage of review). We believe this goes beyond the main scope of this paper.
- Minimum emission level: we agree with the reviewer that "has a small effect" is too vague. We have therefore quantified the main results of the SI 5.2 section, which we included in the previous revision, in the discussion.

[1] Drouet, L., Bosetti, V., & Tavoni, M. (2015). Selection of climate policies under the uncertainties in the Fifth Assessment Report of the IPCC. *Nature Climate Change*, 5(10), 937–940.
<https://doi.org/10.1038/nclimate2721>

[2] Lamontagne, J. R., Reed, P. M., Marangoni, G., Keller, K., & Garner, G. G. (2019). Robust abatement pathways to tolerable climate futures require immediate global action. *Nature Climate Change*, 9(4), 290. <https://doi.org/10.1038/s41558-019-0426-8>

[3] van Vuuren, D. P., van der Wijst, K. I., Marsman, S., van den Berg, M., Hof, A. F., & Jones, C. D. (2020). The costs of achieving climate targets and the sources of uncertainty. *Nature Climate Change*, 10(4), 329-334.

Reviewer #3 (Remarks to the Author):

I will keep this review short, as I have previously reviewed this paper.

I recommend an acceptance of "On the optimality of 2°C targets: a decomposition of uncertainty" by Kaj-Ivar van der Wijst, Andries F. Hof, and Detlef P. van Vuuren. The authors have done a good job of addressing my previous comments. Where they did not make my suggested changes, I believe that their explanations were thoughtful and sufficient.

My only minor suggestion is to rewrite the sentence "A second category consists of models that determine optimal pathways which minimise the costs and benefits of climate policy." Specifically, I

think that the sentence should refer to the "maximization of net benefits" or the "minimization of net costs." I prefer the former as it is more standard, though the latter is also acceptable.

Thank you for this suggestion, it was indeed a wrongly formulated sentence.